# Emerging Developments Regarding Nanocellulose-Based Membrane Filtration Material against Microbes

**DOI:** 10.3390/polym13193249

**Published:** 2021-09-24

**Authors:** Mohd Nor Faiz Norrrahim, Noor Azilah Mohd Kasim, Victor Feizal Knight, Keat Khim Ong, Siti Aminah Mohd Noor, Norhana Abdul Halim, Noor Aisyah Ahmad Shah, Siti Hasnawati Jamal, Nurjahirah Janudin, Muhammad Syukri Mohamad Misenan, Muhammad Zamharir Ahmad, Mohd Hanif Yaacob, Wan Md Zin Wan Yunus

**Affiliations:** 1Research Centre for Chemical Defence, Universiti Pertahanan Nasional Malaysia, Kem Perdana Sungai Besi, Kuala Lumpur 57000, Malaysia; faiznorrrahim@gmail.com (M.N.F.N.); ongkhim@upnm.edu.my (K.K.O.); s.aminah@upnm.edu.my (S.A.M.N.); nurjahirahjanudin@upnm.edu.my (N.J.); 2Department of Chemistry and Biology, Centre for Defence Foundation Studies, Universiti Pertahanan Nasional Malaysia, Kem Perdana Sungai Besi, Kuala Lumpur 57000, Malaysia; aisyah@upnm.edu.my (N.A.A.S.); hasnawati@upnm.edu.my (S.H.J.); 3Department of Physics, Centre for Defence Foundation Studies, Universiti Pertahanan Nasional Malaysia, Kem Perdana Sungai Besi, Kuala Lumpur 57000, Malaysia; norhana@upnm.edu.my; 4Department of Chemistry, College of Arts and Science, Yildiz Technical University, Davutpasa Campus, Esenler, Istanbul 34220, Turkey; syukrimisenan@gmail.com; 5Biotechnology and Nanotechnology Research Centre, Malaysia Agricultural Research and Development Institute, Persiaran MARDI-UPM, Serdang 43400, Selangor, Malaysia; zamharir@mardi.gov.my; 6Wireless and Photonics Network Research Centre (WiPNET), Universiti Putra Malaysia, Serdang 43400, Selangor, Malaysia; hanif@upm.edu.my; 7Research Centre for Tropicalisation, Universiti Pertahanan Nasional Malaysia, Kem Perdana Sungai Besi, Kuala Lumpur 57000, Malaysia

**Keywords:** nanocellulose, membrane filter, microbes, surface functionalization

## Abstract

The wide availability and diversity of dangerous microbes poses a considerable problem for health professionals and in the development of new healthcare products. Numerous studies have been conducted to develop membrane filters that have antibacterial properties to solve this problem. Without proper protective filter equipment, healthcare providers, essential workers, and the general public are exposed to the risk of infection. A combination of nanotechnology and biosorption is expected to offer a new and greener approach to improve the usefulness of polysaccharides as an advanced membrane filtration material. Nanocellulose is among the emerging materials of this century and several studies have proven its use in filtering microbes. Its high specific surface area enables the adsorption of various microbial species, and its innate porosity can separate various molecules and retain microbial objects. Besides this, the presence of an abundant OH groups in nanocellulose grants its unique surface modification, which can increase its filtration efficiency through the formation of affinity interactions toward microbes. In this review, an update of the most relevant uses of nanocellulose as a new class of membrane filters against microbes is outlined. Key advancements in surface modifications of nanocellulose to enhance its rejection mechanism are also critically discussed. To the best of our knowledge, this is the first review focusing on the development of nanocellulose as a membrane filter against microbes.

## 1. Introduction

Throughout the evolutionary process, among the significant issues faced by society today are the protection of natural resources and the implementation of eco-friendly approaches to sustaining a high quality of life. Environmental pollution is a worldwide concern and the majority of pollutants have long-term negative impacts on humans. Focusing on microbial pollution, the most common bulk transportation media for particulate contaminants are air and water. Microbes are microscopic living organisms that can be found everywhere, including in water, soil and air, but they are too small to be seen with the naked eye. These microbes are commonly viruses, bacteria, and fungi and may involve microscopic parasites. Certain microbes are harmful to our health, while others are beneficial. Table 1 shows several types of infectious diseases caused by microbes.

Microbial contamination in water can be dangerous to health, causing serious waterborne disease outbreaks, such as gastroenteritis, cholera, giardiasis and cryptosporidiosis. The most common bacteria involved in these outbreaks are *Shigella dysenteriae*, *Vibrio cholera*, *Legionella* sp., *Escherichia coli*, and *Campylobacter jejuni* [11]. Whereas giardiasis and cryptosporidiosis are gastrointestinal diseases caused by microscopic parasites (protozoa), namely *Giardia duodenalis* and *Cryptosporidium sp.,* respectively. When invading the gastrointestinal tract, these microbes can cause local reactions to their presence and may even cause systemic effects from toxins they secrete (only certain microbes secrete toxins). Some microbes may invade the bloodstream, where they can cause sepsis [12]. Annually, it is estimated that approximately 485,000 people die from diarrheal disease as a result of drinking contaminated water [13]. Hence, microbially contaminated wastewater must be treated before it is discharged into water bodies or water courses.

As mentioned previously, microbes can also be transmitted through the air. According to the World Health Organization (WHO), airborne transmission differs from droplet transmission as it refers to the presence of microbes within droplet nuclei that are typically less than 5 μm in diameter and can circulate in the air for significant periods and be transmitted to others over distances more than 1 m [14]. Whereas droplet transmission occurs when a person is in close contact (within 1 m) with a symptomatic patient with respiratory symptoms such as coughing or sneezing and is thus at risk of exposure to potentially infective respiratory droplets (typically >5–10 μm in diameter). Nowadays, the threat of the newly discovered infectious coronavirus disease (COVID-19) is worrying, as this pandemic outbreak has already killed millions of people worldwide. The outbreak is exacerbated by the occurrence of frequent mutations, which makes it difficult to rapidly produce omnipotent vaccines [1]. Therefore, an effective, robust, and inexpensive air-borne virus removal membrane filter is an urgent need to provide a means to prevent virus spread in hospitals, transportation hubs, schools, and other venues with high social traffic turn-over in order to minimize the risks arising from the COVID-19 pandemic.

Microbe removal can be done through a variety of methods, such as, filtration (either depth filtration or surface screening), partitioning and fractionation (centrifugation), and chromatography (ion-exchange, affinity, gel permeation) [15]. Of these different techniques, filtration is a desirable choice, as it is non-destructive and non-interfering, implying that it will not threaten the quality of biological samples or induce immune reactions. Membrane filters have been made from a variety of synthetic and semi-synthetic polymers, designed to achieve a desired filtration pore size. The membrane filter is also an effective and widely used method for detecting microbiological pollution in collection samples. It requires less planning than certain other conventional methods and is one of the few methods that allows for microorganism separation and subsequent determination. Microbes cannot be retained by the normal membrane filter because the membrane pores are too large. Therefore, it is critical to have a more effective material for microbe filtration, and there are studies that have led to the discovery of new filtering media made from cellulose with efficient filtration capability. The ultimate objective would be to be able to effectively and securely filter microbes from the environment at an affordable cost.

Current filter materials which are typically non-biodegradable and non-renewable have also received much attention among scientists. These membrane filters are primarily made from polymers which include proprietary non-ionic polymers, polytetrafluoroethylene (PTFE), polypropylene (PP), polysulphone (PS), polyvinylidene fluoride (PVDF) and polyethylene (PE) [16,17]. These non-biodegradable polymers when being disposed of after use are known to be harmful to the environment [18]. Figure 1 shows an example of used surgical masks incorporating PP that have been discarded and not properly disposed of thereby causing problems of littering both on land and at sea as well as in waterways. Therefore, scientists urgently need to find better solutions to this problem. It is important to have a more efficient renewable and biodegradable material, of which nanocellulose is a prime candidate.

Cellulose is a versatile industrial product because of its abundance, renewability, biodegradability and ability to be readily chemically modified [19,20,21]. With the development of nanocellulosic materials, since 1977 there has been extensive research into their use in many fields, such as in biocomposites, bioadsorbents, textiles, biomedicals, military, automotives, sensors, energy, packaging, as well as membrane filtration [22,23,24,25,26,27,28,29,30,31]. Generally, nanocellulose can be classified into three types, which are cellulose nanofiber (CNF), cellulose nanocrystals (CNC), and bacterial nanocellulose (BNC) [1,32,33]. The production of nanocellulose can be accomplished in two ways, including up-down or down-up techniques [34]. Up-down techniques can be used to synthesize CNF and CNC. The most popular way to make CNC is to use inorganic acids to acid hydrolyse pure cellulose. To reduce energy consumption, CNF is made via mechanical disintegration with high shear forces, maybe in combination with chemical or enzymatic pre-treatment [35]. On the other hand, bacterial nanocellulose (BNC) is made using a down-up technique, in which cellulose-producing bacteria such as *Acetobacter xylinum* is used [36]. 

According to Hassan et al. (2020) [37], two approaches that utilize nanocellulose for filtration have been explored; namely, the first approach which incorporated them into other polymer matrices to enhance the effectiveness of prepared membranes. This was done by dissolving the polymer in suitable solvents and ensuring that the nanocellulose materials were well dispersed within the polymer solution before being film cast. Meanwhile, the second approach was one first introduced by Ma et al. (2011) [38], in which the authors developed membranes from a nanocellulose layer with adequate porosity laid over polymeric supports without having to dissolve the cellulose matrix and using the film casting method to produce porosity. The latter approach was found to be more desirable and intriguing. 

Nanocellulose-based membrane filters have been found to be effective at removing microbes in previous research [39,40,41]. When compared to synthetic polymers or plastic membranes, a significant benefit of the nanocellulose-based membrane filters is that they are entirely made from natural resources, making their disposal much easier, as they were made up of predominantly biodegradable materials [42]. 

This review is intended to provide a comprehensive overview of the recent advances in the development of membrane filters for microbial removal, which are made up either entirely from nanocellulose or utilizing a modified approach, incorporating nanocellulose. This review will include (1) a description of the types of membrane filters and the rejection mechanism they use; (2) details of the nanocellulose used in the production of membrane filters; (3) a suite of antimicrobial technologies used for nanocellulose functionalization. This manuscript provides knowledge and direction for scientists to stimulate future research in this area.

## 2. Types and Rejection Mechanisms of Membrane Filters

The role of membranes alone in the removal of pathogens is discussed here. Membrane filters can be categorized by the size of the pollutant they are able to reject (see Figure 2), namely: reverse osmosis (0.1–1 nm), nanofiltration (NF) (1–2 nm), ultrafiltration (UF) (2–100 nm) and microfiltration (MF) (100 nm–10 μm). The two most important features of a membrane are its permeability and selectivity. To enhance the productivity of membrane separation processes, it is always necessary to develop membranes with high permeability and high selectivity [43].

Referring to Figure 3, it can be seen that there are various pore sizes of nanocellulose-based membrane filters that are available, depending on their origin. The different pore sizes of nanocellulose membrane filters fulfil various filtration modes. Thus, nanocellulose membrane filters obtained from electrospun CNF usually have a pore size of more than 100 nm (Figure 3a). Thus, making this membrane filter more suitable as an MF. Meanwhile, CNF (Figure 3b) and CNC (Figure 3c,d), which are obtained through other methods usually have a smaller pore size, which ranges from 5 nm to 100 nm. Thus, this makes them suitable for use in NF and UF. The pore size of nanocellulose depends on several factors, such as the cellulose origin, the isolation process concentration and processing conditions as well as pretreatments administered to the cellulose. Moreover, CNF has more advantages to be used as a membrane filter. This is because CNF led to a higher strength and modulus compared to CNC, due to CNF’s larger aspect ratio and fibre entanglement, but lower strain-at-failure because of their relatively large fibre agglomerates [44]. Table 2 summarized the properties of CNF films from different sources.

**Table 2 polymers-13-03249-t002:** Representative studies on the properties of CNF films from different raw materials.

Source of CNF	Modulus (GPa)	Tensile Strength (MPa)	Strain to Failure (%)	Reference
Pulp	10.4–13.7	129–214	3.3–10.1	[45]
Kraft pulp	17	250	2–6	[46]
Wood	6.2–6.9	222–233	7.0–7.6	[47]
Wood	13	223	-	[48]

**Figure 3 polymers-13-03249-f003:**
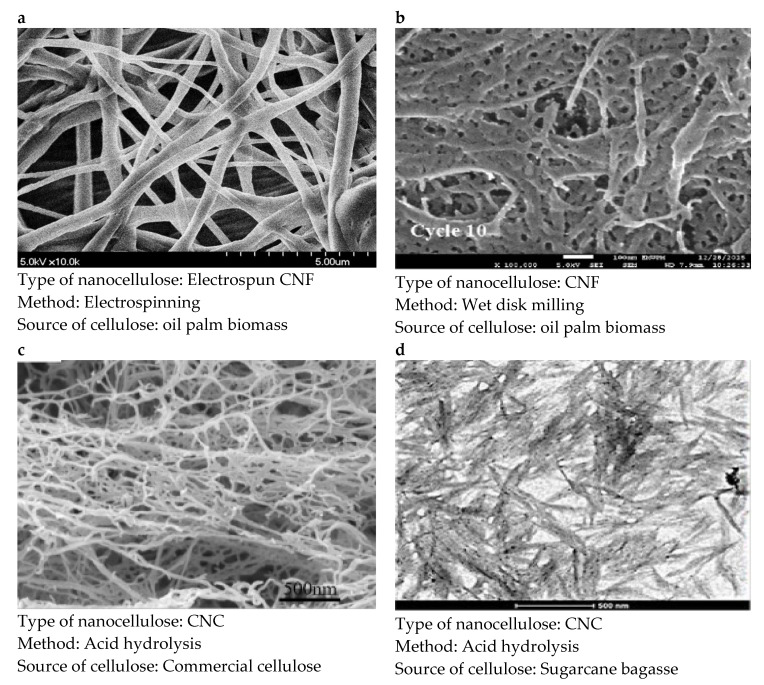
The morphology of nanocellulose produced from various methods. (**a**) electrospinning, (**b**) wet disk milling, (**c**,**d**) acid hydrolysis. This figure is adapted with permission from the [49,50,51,52].

Figure 2 illustrates the membrane filtration spectrum, which operates by utilizing the size exclusion method in rejecting or inhibiting the pathogenic microorganisms. Wu et al. (2019) [53] described it as an established, reliable, and robust method, considering its ability to physically remove various types of infective microorganisms, including virus. Of note, the other method that utilized the affinity principle could also be used for filtration. 

There are two types of membrane filters with different pore sizes, commonly used in the retention of microorganisms. The first one is the MF membrane, which has a pore size of 0.1–10 µm, while the other is the UF membrane, which has a smaller pore size, ranging from 5 to 100 nm. Both types of membrane filters are applicable to the removal of protozoa and algae (i.e., size range between 3 to 14 µm) [54]. In addition, Francy et al. (2012) [55] outlined that tertiary disinfection is not necessary, as the pore size of both MF and UF membranes are too nominal when compared to the size of coliform bacteria, suggesting the total removal of bacteria by size exclusion of the membranes. However, a particular concern was raised regarding the removal of virus via direct membrane filtration considering its smaller size compared to bacteria.

Size exclusion is a widely used technique in the chromatography method, which separates molecules, depending on their sizes or “hydrodynamic volume” in solution. Filtration takes place through a gel composed of spherical beads with a particular size distribution of pores. When molecules of various sizes are incorporated or omitted from the pores inside the matrix, separation occurs. Small molecules diffuse into the pores, slowing their flow through the column, whereas large molecules (or having the greatest hydrodynamic volume) do not penetrate the pores and are eluted in the column’s void volume [56]. As a result, molecules segregate according to their size as they move down the column and are eluted in decreasing order of molecular weight (MW). 

There are a few criteria that influence the effectiveness of the size exclusion technique, particularly the pH and ionic strength of the load buffers, which have a major impact on the retention of diverse specimens. In neutral membranes, the sieving behaviour of charged pollutants is different from that of neutral pollutants. Due to electrostatic interactions with ions in solution, charged pollutants have a double layer of electrical charge on their surface. A solution entering a pore will compress this electrical double layer if the pollutants and pore sizes are of the same magnitude. This is not energetically advantageous, resulting in a reduction in the sieving of the charged pollutants. Interestingly, when an ionic solution encounters a charged membrane, the Donnan model gives a well-known classical description of the electrochemical equilibrium that occurs. It ignores ion size effects while accounting for electrostatic interactions, since the theory regards ions as point charges [57]. When neutral pollutants are applied to a charged membrane, similar effects occur. The presence of neutral pollutants causes a compression of the electrical double layer, associated with the pore wall. With similarly charged pores and pollutants, this impact is amplified much further. Due to charge repulsion, membranes with charged ligands (with identical charge) have poorer sieving of specimens [58]. 

On the other hand, membrane filters utilize the affinity principle, known to use adsorption to remove pollutants based on the electrostatic interaction between functional groups of the membranes with the pollutant. This type of membrane filter includes composite or hybrid filter structures that consist of a porous substrate with either nanocellulose moieties attached to their surface or impregnated within. It is interesting to note that size exclusion and affinity regime approaches have been explored in many studies on membrane filters utilizing the nanocellulose.

Adsorption is an exothermic surface-based process in which molecules of a substance in a certain state aggregate on an adsorbent surface. The adsorbate is the substance that is adsorbed on the adsorbent. Desorption, on the other hand, is the release of adsorbed molecules from the adsorbent’s surface, which is the reverse of adsorption. Adsorption of molecules to the adsorbent surface can take place in two ways: “physical adsorption,” also known as “physisorption,” and “chemical sorption,” also known as “chemisorption.” This is determined by the interactions of the molecules with the surface. Weak forces, such as electrostatic interactions and Van der Waals forces, are involved in physical adsorption. Chemical adsorption results in the formation of strong chemical bonds, such as covalent bonds, between the surface and the adsorbed molecules. A monomolecular layer (monolayer) is developed on the adsorbent surface during chemical adsorption, whereas a thick multilayer is created during physical adsorption on the adsorbent surface [59].

### 2.1. Fabrication of Nanocellulose Membrane

Numerous nanocellulose membrane production processes have been devised, taking into account the unique features of nanocellulose fibres and the membrane casting suspension, as shown in Figure 4 and summarized as follows.

#### 2.1.1. Vacuum Filtration

Vacuum filtration, followed by optional hot-pressing, is a rapid, easy, and accessible procedure for producing layered structures of nanocellulose membrane filters. The amount and concentration of nanocellulose suspensions could be used to alter the thickness and pore size of the resulting membranes [61].

#### 2.1.2. Casting Evaporation and Coating Self-Standing

Normally, self-standing membranes are made by evaporating a dilute nanocellulose suspension in a petri dish. In general, to avoid agglomeration, the nanocellulose dispersion should be diluted to a low concentration (depending on surface chemistry and fibril diameter, but usually less than 1 wt%) [62,63].

#### 2.1.3. Electrospinning 

Electrospun membranes have a smaller base weight, a greater effective surface area, and a higher effective porosity, with pores that are continually interconnected. Nanocellulose could change the membrane surface charge density, increase total effective surface area, and improve functional group density by incorporating it into electrospinning membranes. Furthermore, the nanocellulose content of this multi-layered nanofibrous system could alter the mean pore size and pore size distribution, and hence the separation performance [60].

## 3. Attributes of Nanocellulose Membrane Filtration of Microbes

In filtration systems, diameter, length, cross-section shape, internal structure (cellular or solid), and strength properties, which include tensile strength, stretch or elongation, and stiffness, are the most important physical characteristics of fibres for use in filter media. Fibre qualities that optimize the bulk, air permeability, and pore size of the filter media are ideal. The purpose of the filter design is to optimize bulk and air permeability to allow for breathing while reducing pore size to improve filtration efficiency [64].

Generally, polypropylene (PP) is used to produce the mask accords with the technical standards. The pore diameter of polypropylene is 25 mm. They were treated with dimethyldioctadecylammonium bromide to impart a positive electrical charge capable of attracting bacteria. Bacterial or viral filtration efficiency was almost 100% for the PP mask [65].

Normal membrane filters usually have pores that are too large to retain microbes. The advances in nanotechnology have made nanocellulose the more suitable material for the filtration of microbes. Nanocellulose with pore dimensions measuring between 1 and 100 nm offers certain unique characteristics, which include high strength, chemical inertness, hydrophilic surface chemistry and high surface area, thereby making it a promising material for use as a high-performance membrane filter that can effectively remove microbes from either air or liquids [22,49,66,67,68,69,70,71,72]. Moreover, membrane filter constructed entirely of CNF has recently been discovered to be capable of filtering even the tiniest viruses with up to 99.9980–99.9995% effectiveness. 

The comparison between the PP membrane and nanocellulose membrane is tabulated in Table 3. On the other hand, Table 4 summarizes the importance of several special properties of nanocellulose which are related to the application as a membrane filtration material against microbes.

**Table 3 polymers-13-03249-t003:** Comparison of PP and nanocellulose membrane.

Characteristic	PP	Nanocellulose
Fibre length (nm)	-	400
Diameter (nm)	25,000	1–100
Efficiency against pathogens	~100%	99.9980–99.9995%
Tensile modulus (GPa)	1.5	145
Tensile strength (GPa)	0.02	7.5
Poison’s ratio	0.4	0.3

**Table 4 polymers-13-03249-t004:** Certain interesting properties of nanocellulose related to membrane filtration materials.

Property	Advantages	Reference
Nanoporosity	Good virus filtration using size-exclusion method. Typically, the pore size of nanocellulose is below 100 nm.	[15]
Surface functionalization	Functionalization nanocellulose with several compounds to make it cationic charged causes an increase in its binding affinity towards viruses.	[73]
High specific surface area	Provides a large surface area for functionalization. Thereby increasing interaction efficiency.	[74]
Renewable	Nanocellulose can be easily sourced from plant bio-waste. Its use can eliminate the use of other non-renewable polymers as mentioned in the Introduction section.	[49,66]
Biodegradability	An important aspect to save the environment. It is biodegradable in landfills. Hence, current environment issues from used and discarded surgical masks can be reduced or even eliminated.	[75]
High mechanical strength	High strength membrane filters can be fabricated using it.	[76]
Stability in water	It can reduce biofouling of membrane filters. This is important for application in membrane filters for wastewater.	[77]

As described earlier, nanocellulose can be classified into three types (CNF, CNC, and BNC) according to their manner of origin. These three types are as shown in Table 5, below. Essentially cellulose is a molecule that consists of β-1, 4-glucose, and three active hydroxyls at the C2, C3, and C6 sites of the glucopyranose ring, and its configuration provides sufficient sites for several surface functionalization’s. These sites may undergo oxidation, esterification, and etherification to enable these variable functional groups that may include aldehyde groups and quaternary ammonium. The details of these surface functions will be explained in the following section.

Each type of nanocellulose has different dimensions and properties, mainly due to their different methods of preparation/fabrication [78]. There are various methods used to extract cellulose nanoparticles which then result in these particles having different crystallinities, surface chemistries and mechanical properties [79]. These production methods range from top-down enzymatic/chemical/physical methods aimed at isolating them from wood and forest or agricultural residues to bottom-up production of cellulose nanofibrils from glucose by bacterial action [80].

**Table 5 polymers-13-03249-t005:** Types of nanocellulose according to their sources, treatments and dimensions [60,81,82,83].

Nanocellulose	Abbreviation	Sources	Main Treatments	Dimensions
Cellulose nanofiber 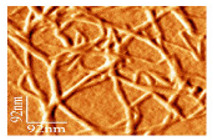	CNF	Plants	Mechanical fibrillation	Diameter: 5–50 nm Length: Several µm
Cellulose nanocrystals nanowhiskers/nanorods 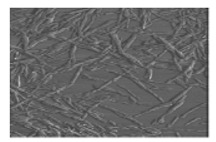	CNC	Plants	Acid hydrolysis	Diameter: 2–20 nm Length: 100 nm to several µm
Bacterial nanocellulose/biocellulose 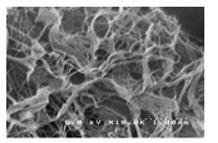	BNC	Microorganisms	Polymerization and crystallization	Diameter: 2–4 nm Length: 100µm

## 4. Modifications on Nanocellulose to Improve Filter Efficiency

The surface functionalization of nanocellulose is a key step to promoting an increase in the efficiency of a membrane filter. This is an important step when the membrane filter operates using the affinity regime. This can be done using different strategies of surface functionalization that will involve the chemistry of hydroxyl function [84]. Referring to Figure 5, a search using keywords ‘functionalization of nanocellulose’ was performed by lens.org, (https://www.lens.org/; accessed on 15 February 2021), a problem-solving non-profit social enterprise website utilizing linked open knowledge artefacts and metadata. It was found that manuscripts focusing on the functionalization of nanocellulose have been increasing from 2011 until now. This showed that the area of research interest has grown among scientists in this decade. 

The surface functionalization is also important to increase the performance and lifetimes of the nanocellulose-based membrane filter. Nanocellulose is a fibrous water-loving polymer, due to the presence of an abundant number of OH groups. Thus, it can cause the membrane filter to swell and weaken in the presence of water. However, surface functionalization can overcome this problem by enhancing the hydrophobicity of nanocellulose [71,72] Besides that, surface functionalization also can improve the mechanical strength of nanocellulose-based membrane filters [85].

Surface functionalization on nanocellulose can be made through several processes, such as oxidation, esterification, and etherification, which eventually results in the introduction of new functional groups on the material. Other than this, previous studies have showed that nanocellulose can also be subjected to modification by adding compounds such as aldehyde, quaternary compounds (both cationic and anionic forms), activated carbon, citric acid, antibiotics, and nanomaterials. For instance, the aldehyde groups are grafted onto nanocellulose through the oxidation process, using oxidants such as periodate sodium and 2,2,6,6-teramethylpiperidinyloxy (TEMPO), which results in the TEMPO positioning on the surface of the nanocellulose under aqueous conditions, while the hydroxyl group located at the C6 position of the nanocellulose can be converted to carboxyl and aldehyde functional groups. 

Besides this, low toxicity and environmentally friendly quaternary compounds, such as poly(N,N-dimethylaminoethyl methacrylate), amines, anionic polyelectrolytes, and polyglutamic acid could be used to quaternize nanocellulose to improve its efficiency as a membrane filter, as these quaternary compounds have the ability to form electrostatic affinity towards microbes. This quaternization process can be performed using grinding and high-pressure homogenization processes. Table 6 illustrates several examples of functionalized nanocellulose using quaternary compounds for virus removal applications. Most viruses and certain microbial species have polar charges on their surface; thus, modification on the surface charge of the nanocellulose improves the electrostatic interaction properties of the material, which consequently results in high efficiency filtration.

The main challenge with chemical functionalization to nanocellulose is to select an appropriate time for the functionalization to occur. Surface functionalization can be carried out during the preparation step or post-production of nanocellulose [86]. This process can be greatly affecting the final properties of functionalized nanocellulose such as crystallinity, yield, dimensions and morphology, surface chemistry, physicochemical, and thermal properties. If the wrong selection of functionalization approach, the 3D crystal network can be interrupted during modification, thus the mechanical properties of nanocellulose could deteriorate, which could consequently limit the applications of modified nanocellulose. In some approaches, nanocellulose is modified during the production step [87], while in other strategies the nanocellulose was produced, first followed by the modifications [88,89,90].

For example, Henschen (2019) [91] functionalized the nanocellulose with polyelectrolytes. It was found that the suspended nanocellulose is too small to be easily recovered if added to different solutions and it has a tendency to aggregate in polyelectrolyte solutions. Therefore, to adsorb polyelectrolytes onto nanocellulose, the functionalization is preferably done after the production of nanocellulose.

In the following section, several interesting findings concerning quaternized nanocellulose for microbial removal will be discussed.

## 5. Recent Developments on Nanocellulose as a Filtration Material against Microbes

In this section, several developments concerning nanocellulose-based membrane filters capable of removing microbes will be reviewed. An important aspect of the modification of nanocellulose materials is to increase the binding affinity of the materials towards microbes. There are a number of studies that focused on the filtration removal of viruses and bacteria; however, very limited studies have been conducted, which concern other types of microbes, such as fungi, algae and protozoa. 

### 5.1. Viruses

The ensnarement of viruses is one of the most crucial steps in biopharmaceutical and clinical processes and applications [92]. Of the various types of microbes, virus is among the smallest and most difficult to deal with, as compared to other microbes. 

Exploration of nanocellulose as a filtration material against several types of viruses has received much research attention. As mentioned earlier, several viruses, including COVID-19, are airborne viruses that can be dispersed and spread through human nasal or saliva secretions from an infected person. Therefore, in order to minimize infection risks from viruses, an efficient, robust and affordable air-borne virus removal filter is an urgent requirement. Multiple research articles were recently published with regard to this type of air filter. 

Several factors, such as filter thickness, pore size, number of layers, size of the virus, the charge on the filter surface, its ionic strength and surface chemistry are usually influenced by the efficiency of the air filtration process [15]. Generally, the use of size-exclusion type filtration has several benefits, such as flexibility and ease of use since it provides virus removal predictability through its physical properties, allows for the filtration of viral markers, enabling easy validation of the filtration process, and does not use toxic or mutagenic chemicals for viral inactivation [15,93,94]. 

Gustafsson et al. (2018) [95] evaluated membrane filter made from nanocellulose in a mille-feuille arrangement of varying thicknesses using a simulated wastewater matrix to explore its ability to remove viruses for drinking water purification applications. The filtrations of various samples of simulated wastewater with its total suspended solid content being 30 nm latex particles as surrogate waste material and 28 nm ΦX174 bacteriophages as the viral contamination. The authors examined the performance of these membrane filters at a pressure of 1 and 3 bar with varying thicknesses of 9 and 29 µm. The data they obtained demonstrate that a membrane filter made from 100% nanocellulose has the capacity to efficiently remove even the smallest of viruses, with up to 99.9980–99.9995%.

Manukyan et al. (2019) [96] fabricated nanocellulose-based mille-feuille type membrane filter for use in upstream applications for serum-free growth media filtration and it was designed to remove ΦΧ174 bacteriophages. The filter performance was evaluated based on its ability to filter small–medium-sized viruses using varying thicknesses of the fabricated membrane filter (i.e., 11 and 33 µm), as well as by varying the operating pressures (i.e., 1 and 3 bar). Based on their results, the 33 µm thick filter showed more stability and had better virus removal as compared to the 11 µm thick filter, although their flux was nominally lower. The findings of this study suggest that the nanocellulose membrane filter would be a viable alternative for the filtration of large volumes of cell culture media in upstream bioprocessing. 

Asper et al. (2015) [97], in their study, used a membrane filter composed of 100% CNF to remove xenotropic murine leukaemia virus (xMuLV). It was found that the particle retention properties of the nanocellulose membrane filter were verified following the filtration of 100 nm latex beads, as shown in Figure 6. The results of this filtration of xMuLV suggested that the nanocellulose membrane filter was useful for removal of endogenous rodent retroviruses and retrovirus-like particles during the production of recombinant proteins. 

Metreveli et al. (2014) [15] reported that one of the most challenging tasks for designing the virus removal membrane filter is tailoring the membrane upper pore size cut-off so that the filter retains viruses with a particle size of between 12 and 300 nm, while allowing for the unhindered passage of proteins which typically range between 4 and 12 nm in size. Therefore, high porosity of the nanocellulose-based filters is required to circumvent the problem of low permeance. In their study, the developed nanocellulose membrane filter, sized at 70 μm with a total porosity of 35%, was able to remove Swine Influenza A Virus (SIV), which had a particle size of 80–120 nm. The latex beads and SIV particles are observed as stacked structures on the surface of the porous membrane filter. They also found that the proteins pass unhindered through the membrane filter. Therefore, the pore size distribution presented in their work is promising for virus filtration applications, especially for large viruses ≥50 nm.

Besides this, Mautner et al. (2021) [98] also produced BNC membrane filters with high porosity for optimized permeance and rejection of nm-pollutants. The BNC was treated with organic liquids (alcohol, ketone, ether) before being further processed into the membrane filter. The treated BNC membrane filter has a porosity of 67%, which is higher than the untreated BNC membrane filter (33%). It also exhibits 40 times higher permeance, caused by a lower membrane density. Despite their higher porosity, the developed membrane filter also still has pore sizes of 15–20 nm, which is similar to the untreated BNC membrane filter. Thus, the developed membrane filter enables the removal of viruses by a size-exclusion mechanism at high permeance.

The strength of the nanocellulose is also important in designing a good membrane filter to remove viruses via a size exclusion mechanism. A study by Quellmalz and Mihranyan, (2015) [85] found that the citric acid cross-linked nanocellulose-based membrane filter has better mechanical performance than the untreated nanocellulose. It was observed that the untreated nanocellulose membrane filter was readily cracked at pressure gradients above 15 kPa, which could be limiting for its industrial application. The improved strength of the cross-linked nanocellulose membrane filter enables increasing the pressure gradient applied for filtration without compromising the integrity of the filter. It is concluded that citric acid cross-linking of nanocellulose is beneficial to be used in several industrial applications for removing viruses.

Previous studies on the surface modification of nanocellulose have led to the improvement of filtration efficiency against viruses. Electrostatic interaction between nanocellulose and viruses is improved dramatically with the incorporation of quaternary compounds as discussed in Section 4, above. For instance, viruses such as coronavirus have a negatively charged surface and would interact with the cationic or anionic charge of nanocellulose-quaternary compounds [99]. Figure 7 shows a schematic diagram of the coronavirus structure with proteins embedded in its bilayer membrane and negatively charged lipid head groups protruding to the outer side of the membrane.

The entrapment of the virus onto nanocellulose matrix is due to the presence of electrostatic force attraction between the negatively charged virus particle and the positively charged nanocellulose membrane. Several studies have demonstrates successful results in filtering negatively charged viruses using cationic nanocellulose. For example, Mi et al. (2020) [101] developed a filtration setup using modified CNC with a positively charged guanidine group to adsorb porcine parvovirus and Sindbis virus and to completely filter out those viruses from water. It is interesting to point out that this filtration system has exceeded the Environment Protection Agency (EPA) virus removal standard requirement for portable water. In addition to the electrostatic interaction between the virus and guanidine group, Meingast and Heldt (2020) [102] outlined that the complete virus removal from water was also due to the protonated guanidine groups on the cationic CNC forming ionic and hydrogen bonds with the proteins and lipids on the virus surface.

Other than that, Rosilo et al. (2014) [103] in their study observed a very high affinity binding between the cationic CNC (known as CNC-g-P(QDMAEMA)s) mixture and cowpea chlorotic mottle virus (CCMV) and norovirus-like particles in water dispersions. Of note, this cationic CNC mixture was prepared by surface-initiated atom-transfer radical polymerization of poly(N,N-dimethylamino ethyl methacrylate) and its subsequent quaternization of the polymer pendant amino groups. 

In addition, the anionic CCMVs could also be removed using functionalized lignin with a quaternary amine. In their study, they found that the CCMVs would form agglomerated complexes with cationic lignin [104]. Therefore, suggesting its potential use as material in the development of membrane filter for the removal of CCMVs.

Besides that, Sun et al. (2020) [105] reported that covalent modification on CNF (i.e., functionalization of nanocellulose) using polyglutamic acid (PGS) and mesoporous silica nanoparticles (MSNs) resulted in the successful filtration of EV71 virus and Sindbis virus. This is particularly due to the interaction between two exposed positively charged amino acids (His10 and Lys14) and the negatively charged MSNs on the modified CNF [105]. 

Table 7 summarizes the development of nanocellulose-based membrane filtration material for virus removal that have been discovered/explored so far. In addition to the guanidine groups, lignin, nanoparticles, and citric acid, nanocellulose could also be functionalized with several other compounds, such as small organic molecules, porphyrin dendrimers and others polymers in order to make it positively or negatively charged [73]. However, it is important to note that not all of these examples have been tested as a filter to remove viruses. It can be summarized that several present studies have shown the capability of nanocellulose as a filtration material for virus removal. Separation by size exclusion and adsorption mechanism are the most common approaches. Factors such as pore size distribution, porosity, thickness, strength and surface functionalization of nanocellulose can greatly influence the filtration efficiency. 

### 5.2. Bacteria

The development of nanocellulose as a filtration material against bacteria also been widely discovered. Generally, the diameter of most waterborne bacteria is larger than 0.2 μm [38]. Thereby, it would be easy for nanocellulose-based membrane filters to entrap most bacteria species using the size-exclusion mechanism. Moreover, as discussed in Section 4 earlier, the modification of nanocellulose by surface functionalization can also be performed to increase the removal efficiency of bacteria. In this review, we highlight several findings concerning bacterial removal using nanocellulose based membrane filters.

Wang et al. (2013) [107] demonstrated that a multi-layered nanofibrous microfiltration system with high flux, low-pressure drops and high retention capability against bacteria (*Brevundimonas diminuta* and *Escherichia coli*) was possible by impregnating ultrafine CNF into an electrospun polyacrylonitrile (PAN) nanofibrous scaffold supported by a poly (ethylene terephthalate) (PET) non-woven substrate. The CNF was functionalized prior to impregnation with carboxylate and aldehyde groups using TEMPO oxidation. It was observed that this CNF-based microfiltration membrane exhibited full retention capability against those bacteria.

Otoni et al. (2019) [108] developed a cationic CNF compound using Girard’s reagent T (GRT) and shaped it into foam using several protocols, such as cryo-templating to remove the ubiquitous human pathogen *Escherichia coli*. The porosity of this foam, which is associated directly with its surface area and pore size plays a significant role in the removal of *Escherichia coli*. The cryogel foams produced by this method had porosities of *circa* 98% and were established to be able to achieve an approximately 85% higher anti *Escherichia coli* activity when compared to sample foams made up of unmodified CNF. The cationic CNF using GRT demonstrated good potential for both air and liquid filtration, with excellent absorbency through functional coating. Access to safe drinking water in high- and low-income countries has become one of the biggest challenges in the world as natural resources become scarcer. 

Gouda et al. (2014) [109] invented a modified electrospun CNF containing silver nanoparticles (AgNPs) as a water disinfecting system for water purification systems. The AgNP content, physical characterization, surface morphology and antimicrobial efficacy of the developed membrane filter were then studied. AgNP, which belongs to the group of biocidal nanoparticles, has antimicrobial properties and is commonly used due to its size quantization effect. This can cause a shift in the reactivity of metals in the nanoscale. The developed membrane filter had an excellent ability to remove bacteria, including *Escherichia coli*, *Salmonella typhi*, and *Staphylococcus aureus* with a percentage filtration of more than 91% in contaminated water samples. 

Ottenhall et al. (2018) [110] developed a CNF-based membrane filter, modified with polyelectrolyte multilayers to produce multilayer cationic polyvinyl amine (PVAm) and anionic polyacrylic acid (PAA). The authors successfully modified the CNF with cationic polyelectrolyte PVAm, together with the anionic polyelectrolyte PAA in either single layers or multilayers (3 or 5 layers) using a water-based process at room temperature. Based on filtration analysis, the functionalized CNF-based membrane filters with several layers were physically able to remove more than 99.9% of *Escherichia coli* from water. The three-layer membrane filter could remove more than 97% of cultivatable bacteria from natural water samples, which was a remarkable performance, as compared with the simple processing technique using plain nanocellulose filters. 

Table 8 summarizes the effectiveness of nanocellulose-based membrane filters that have been functionalized with bioactive compounds for the removal of bacteria. It can be concluded that bacterial separation by size exclusion mechanism is easier as compared to the virus. This is because the size of bacteria is usually larger as compared to a virus. The surface functionalization on nanocellulose is capable of introducing anti-bacterial properties to the developed filtration material. However, limited studies were reported for the removal of other bacteria species using nanocellulose-based filtration material.

### 5.3. Other Types of Microbes

Nanocellulose would also be able to act as a removal agent for other types of microbes which are larger in size than bacteria, such as fungi, algae and protozoa. However, it is noteworthy that there is still a lack of studies regarding this matter. To the best of our knowledge, there are no available reports on the development of a nanocellulose-based membrane filter for the removal of fungi. 

Algae is also a major contributor to microbial contamination in water resources and their presence could change the taste or odour of water. For example, blue–green algae and coloured flagellates (especially the *Chrysophyta* and *Euglenophyta* genera of algae) are the best-known algae that cause contamination in water resources. Furthermore, green algae may also be a significant contamination factor as well [117]. Hence, the potential of nanocellulose should be explored by scientists to define their role as a membrane filtration material suitable for removing algae and protozoa from the contaminated water efficiently. Algae and protozoa are known to have a larger size than viruses and bacteria, thus the removal of these microbes could be effectively carried out using the size-exclusion mechanism. 

However, similar to viruses and bacteria, the nanocellulose needs to be modified with other compounds such as metal nanoparticles, enzymes and proteins in order to increase its filtration efficiency [118]. Studies have shown that different charges between the cellular membrane of algae and protozoa do play a dominant role in the adsorption/retention of these microbes on a filtration membrane’s surface (i.e., through the electrostatic interaction principle) [110,119]. 

A previous study carried out by Ge et al. (2016) [120] discovered the sustainability and the most efficient approach in harvesting algae using a modified CNC. The modification was made by introducing a 1-(3-aminopropyl)-imidazole (APIm) structure as a reversible coagulant. As shown in Figure 8, the coagulation process occurs when the positively charged CNC–APIm interacts with the negatively charged *Chlorella vulgaris* in the presence of carbon dioxide (carbonated water). Their findings are in agreement with the works of Qiu et al. (2019) [121], in which it was found that harvesting efficiency could reach up to 85% with only 0.2 g CNC–APIm mass ratio, 5 s of CO_2_ sparging time, and a 50 mL/min flow rate. This signifies that the CNC–APIm complex could be an alternative to current conventional coagulants for harvesting algae in industrial applications.

Algae harvesting is important for biodiesel industry and many studies have been carried out to increase its sustainability on a global scale. For example, the capability of CNF and CNC in harvesting algae was investigated by Yu et al. (2016) [122]. In their study, they discovered that the CNF did not require any surface modification to harvest the algae, as it played a role as an algae flocculant via its network geometry, something that the CNC (even cationic modified CNC) could not do. By referring to Figure 9, flocculation of algae did not happen when CNC was used, as the freely moving algae cannot be entrapped by the nanoparticle structure formation of CNC. However, this study only focuses on the flocculation capability of CNF and CNC, which could lead to a further study on the filtration efficiency of both materials for algae harvesting. This finding could indirectly point to the development of a nanocellulose-based membrane filter for algae removal in the future.

Overall, nanocellulose has shown its capability to filter algae. The functionalization is also important to improve filtration efficiency. However, the development of nanocellulose as a filtration material of fungus and protozoa is still limited. It is important to further investigate the capability of nanocellulose to filter these microbes. Moreover, several other factors which could influence the filtration efficiency, as discussed before, can also be considered for future studies.

## 6. Challenges and Future Recommendations

This review has shown several undoubtedly interesting features of nanocellulose which is useful in filtering viruses, bacteria, fungi, algae, and protozoa through the mechanism of membrane filtration. The properties and characteristics of nanocellulose as a filtration material is very promising and is an exciting area for current and future research. Several recent developments in the application of nanocellulose as a membrane filtration material were discussed here. It is interesting to note that the functionalization of nanocellulose with a variety of functional groups is among the important key factors of success to enhance the removal of microbes from air and water.

Even though different works on the nanocellulose as a membrane filter material has shown several effective findings, improvements in this area are still needed. There are several other types of microbial species which have not been explored. The use of nanocellulose-based membrane filters as a means of eradicating the COVID-19 virus has also not been explored. Moreover, research on the use of nanocellulose-based membrane filter materials to remove fungus, protozoa and algae is still very limited. Therefore, more works concerning those microbes remains an urgent need. 

The functionalization of nanocellulose is a very important step to obtaining the improved performance of membrane filtration material. In this review, several compounds have been shown to be capable of being incorporated with nanocellulose. However, their side effects towards the environment as a result of this functionalization of nanocellulose is also an important consideration. For example, the functionalization of nanocellulose with TEMPO can be harmful to the environment. This is because the synthesis of TEMPO can generate chemical by-products which are toxic to aquatic life when released as waste effluent into the environment [123]. In the future, research should also be focused on this concern to determine the actual feasibility and sustainability of these developed functionalized nanocellulose products towards the environment. 

In addition, further research on generating hybrid nanostructures on the surface of nanocellulose to enable interaction with multiple species of microbes is urgently needed, which will pave the way towards the development of new materials capable of eliminating various kind of microbes at once.

This review has identified several difficulties concerning the use of nanocellulose as a membrane filter material. One of them is particularly related to the high production cost especially in large industrial scale. Furthermore, high energy consumption during the production of nanocellulose is still a concern and cries for a more reliable and reproducible production technique to pave way to using nanocellulose as a commercially viable membrane filter material. To the best of our knowledge, there is much progress in research studies focusing on reducing the energy consumption and production cost of nanocellulose and have been attempted in numerous pilot-scale productions worldwide. Other than that, issue regarding the biodegradability of nanocellulose as adsorbent need to be evaluated by considering several factors such as the types of water and presence of certain microbes that may cause cellulose degradation. 

All in all, nanocellulose is a good alternative for a membrane filter material and is expected to be fully utilized in numerous industries in the near future, considering the solutions for the outlined challenges and difficulties that have been met.

## Figures and Tables

**Figure 1 polymers-13-03249-f001:**
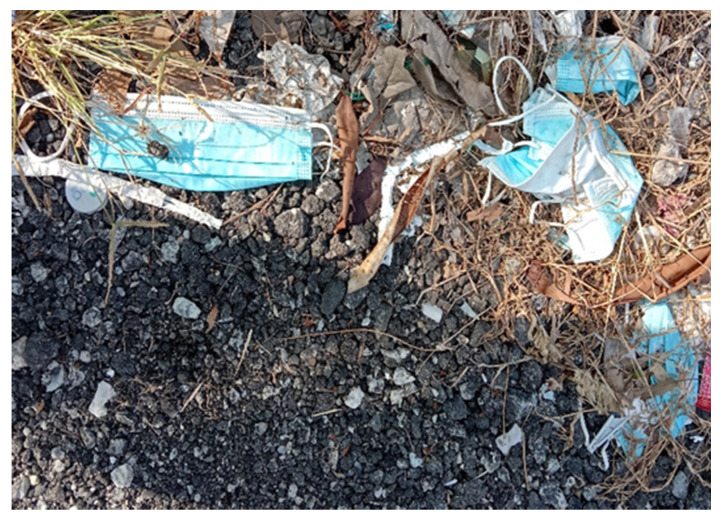
Used surgical masks that were not properly disposed.

**Figure 2 polymers-13-03249-f002:**
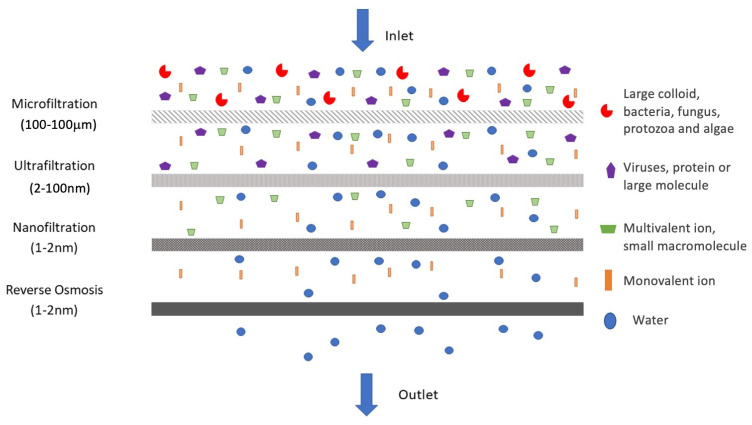
Comparison of the size and type of contaminants rejected by membrane filtration techniques.

**Figure 4 polymers-13-03249-f004:**
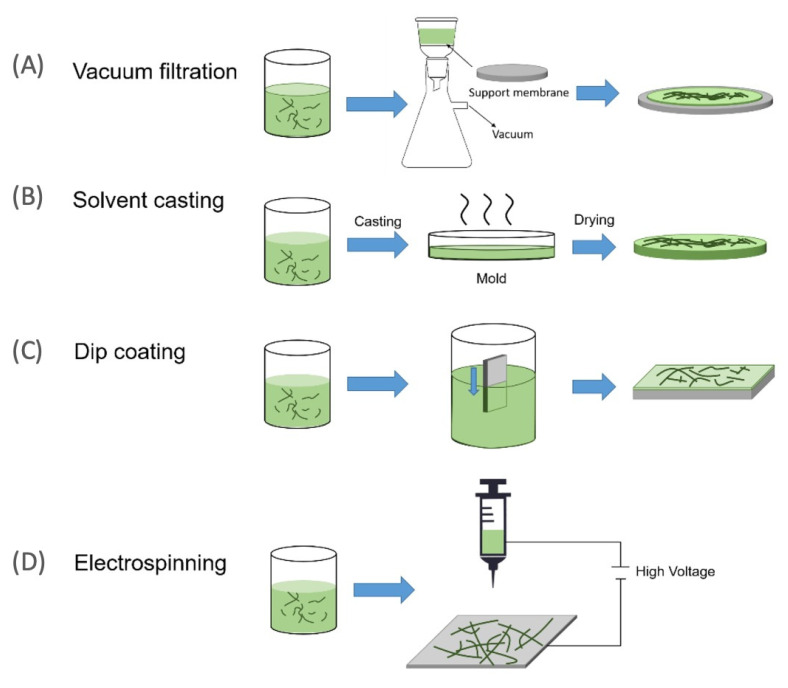
Various methods used for the fabrication of the nanocellulose membrane filter. Reproduced with permission from [60].

**Figure 5 polymers-13-03249-f005:**
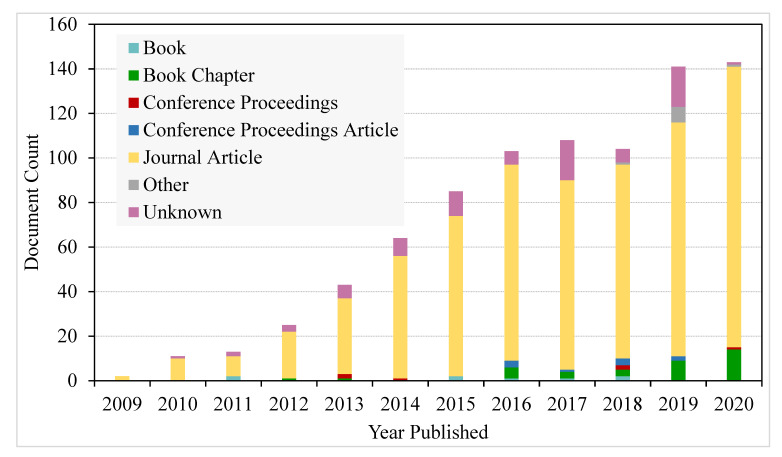
A chart of published manuscripts focusing on the functionalization of nanocellulose from Lens.org.

**Figure 6 polymers-13-03249-f006:**
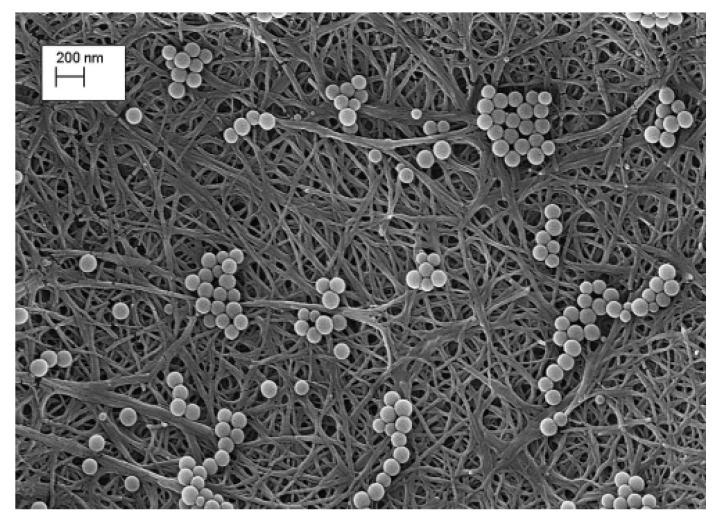
SEM images of 100 nm latex beads retained on the nanocellulose membrane filter. Reproduced with permission from [97].

**Figure 7 polymers-13-03249-f007:**
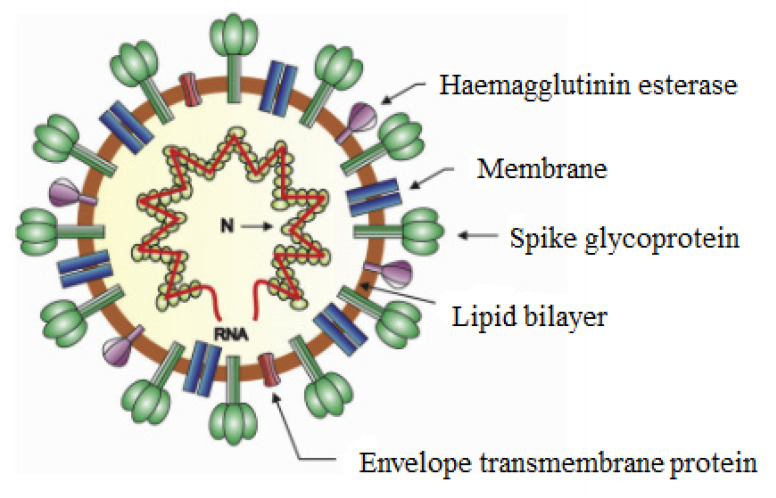
A structure design of coronavirus particle. Reproduced with permission from [100].

**Figure 8 polymers-13-03249-f008:**
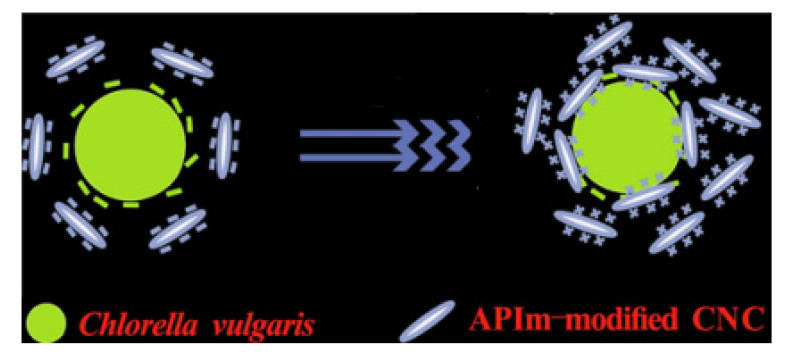
Illustration on the electrostatic attraction between *Chlorella vulgaris* and APIm-modified CNC. This figure was adapted from [120].

**Figure 9 polymers-13-03249-f009:**
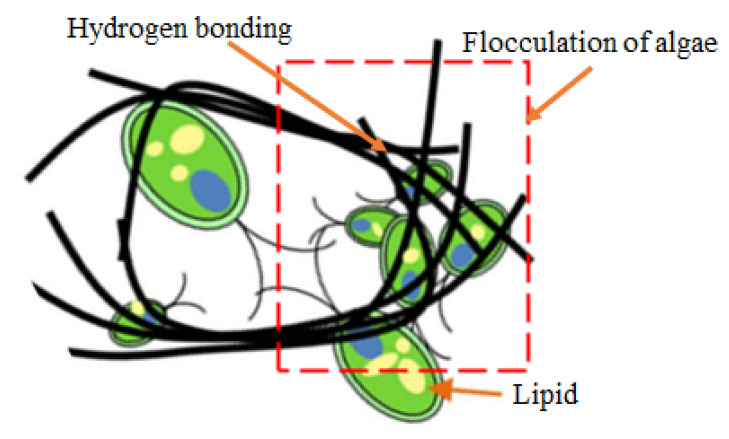
Schematic design of CNF induced microalgal flocculation. This figure was adapted from [122].

**Table 1 polymers-13-03249-t001:** Several infectious diseases caused by microbes.

Infectious Disease	Microbe That Causes the Disease	Type of Microbe	Reference
Coronavirus (COVID-19)	Severe acute respiratory syndrome coronavirus 2 (SARS-CoV-2)	Virus	[1]
Cold	Rhinovirus	Virus	[2]
Chickenpox	*Varicella zoster*	Virus	[3]
German measles	Rubella	Virus	[4]
Whooping cough	*Bordatella pertussis*	Bacteria	[5]
Bubonic plague	*Yersinia pestis*	Bacteria	[6]
TB (Tuberculosis)	*Mycobacterium tuberculosis*	Bacteria	[7]
Malaria	*Plasmodium falciparum*	Protozoa	[8]
Tinea barbae (dermatophyte infection)	*Trichophyton rubrum*	Fungus	[9]
Athletes’ foot	*Trichophyton mentagrophytes*	Fungus	[10]

**Table 6 polymers-13-03249-t006:** Functionalized nanocellulose with quaternary compounds for virus removal applications.

Functional Group	Chemical Structure
a. Functional group: aminoethyl methacrylate or poly(*N*-(2-aminoethylmethacrylamide)	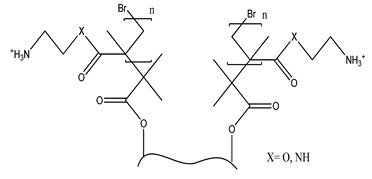
b. Functional group: 2,3-epoxypropyl trimethylammonium chloride	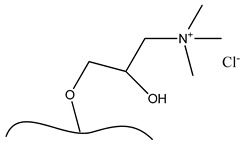
c. Imidazolium	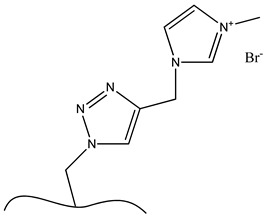
d. Pyridinium	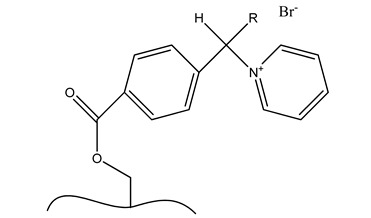
e. e-vinylpyidine	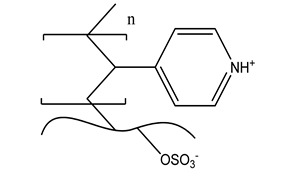
	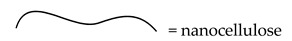

**Table 7 polymers-13-03249-t007:** Nanocellulose developed filtration material for virus removal.

Microbes	Type of Nanocellulose	Functionalization	Findings	Reference
A/swine/Sweden/9706/2010 (H1N2)—Swine influenza	BNC	Not applicable	The newly developed non-woven, μm thick membrane filter consisting of crystalline BNC able to remove virus particles solely based on the size-exclusion principle, with a log 10 reduction value ≥6.3, thereby matching the performance of industrial synthetic polymer virus removal filters currently in use.	[15]
Xenotropic murine	BNC	Not applicable	The developed BNC membrane filter could remove the endogenous rodent retroviruses and retrovirus-like particles.	[97]
MS2 viruses	BNC	Not applicable	This study highlights the efficiency of the nanocellulose-based membrane filter in removing/filtering out the ΦX174 bacteriophage with value of 5−6 log virus clearance (28 nm; pI 6.6).	[53]
ColiphagesΦX174	BNC	Not applicable	The nanocellulose-based membrane filter exhibited 5−6 log virus clearance of MS2 viruses (27 nm; pI 3.9). This study also showed the possibility of producing cost-efficient viral removal filters (i.e., manufacturing process).	[53]
Parvoviruses	BNC	Not applicable	The developed filter was the first non-woven, wet-laid membrane filter composed of 100% native cellulose. This study showed that the non-enveloped parvoviruses could be eliminated using this filter.	[106]
EV71	CNF	Polyglutamic acid and mesoporous silica nanoparticles	This study showed that the modified microfibers could strongly adsorb the epitope of the EV71 capsid which is useful for virus removal.	[105]
Sindbis virus	CNC	Guanidine	Functionalization of guanidine on CNC resulted in over 4 log removal value against the Sindbis virus.	[101]
Porcine parvo virus	CNC	Guanidine	Authors also revealed that functionalization of guanidine on CNC managed to remove the Porcine parvo virus with over 4 log removal value.	[101]

**Table 8 polymers-13-03249-t008:** Nanocellulose developed filtration material for bacterial removal.

Microbes	Type of Nanocellulose	Functionalization	Findings	Reference
*Escherichia coli*	CNC	Silver nanoparticles	It possesses high adsorption capacity and is reusable. Beneficial in total removal of *Escherichia coli*.	[40]
*Bacillus subtilis* and *Escherichia coli*	CNF	ZnO and CeO_2_	It has high anti-bacterial activity, MIC_50_ against Bacillus *subtilis* (10.6 µg mL^−1^) and *Escherichia coli* (10.3 μg mL^−1^).	[111]
*Escherichia coli*	BNC	Not applicable	The significance of Brownian motion caused by microorganisms captured with BNC-based membrane filter through theoretical modelling and filtration experiments was investigated by the authors. It was found that the BNC-based filter was capable of filtering the bacteria.	[112]
*Escherichia coli*, *Staphylococcus aureus*	CNF	Activated carbon	The two-layer AC/OCNF/CNF membrane able to remove *Escherichia coli* bacteria up to ~96–99% and inhibits the growth of *Escherichia coli* and *Staphylococcus aureus* on the upper CNF surface	[41]
*Escherichia coli*	BNC	Silver nanoparticle	Higher amount of silver nanoparticles loaded onto the BNC membrane surface could increase the inhibition zone hence highlighting its good antimicrobial property against *Escherichia coli*.	[113]
*Escherichia coli, Staphylococcus aureus, Pseudomonas aeruginosa*	BNC	Silver nanoparticle	BNC-silver nanoparticle membrane showed strong antimicrobial activity against Gram positive (*Staphylococcus aureus*) and Gram-negative (*Escherichia coli* and *Pseudomonas aeruginosa*) bacteria.	[114]
*Escherichia coli, Staphylococcus aureus*	BNC	Silver nanoparticle	The developed Ag/BNC membrane exhibited good property as antimicrobial agent against *Escherichia coli* and *Staphylococcus aureus* as the antibacterial ratio against *Escherichia coli* and *Staphylococcus aureus* reached 99.4% and 98.4%, respectively.	[115]
*Escherichia coli*	CNF	Polyethersulfone (PES) membranes	TEMPO oxidized-CNF coating is effective against *Escherichia coli*. The effectiveness was attributed to the pH reduction effect induced by carboxyl groups	[116]

## Data Availability

Not applicable.

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
