# Peer review of "Emerging Developments Regarding Nanocellulose-Based Membrane Filtration Material against Microbes"

_polymers, 2021, doi:10.3390/polym13193249_

Round 1
Reviewer 1 Report
This review cites many literatures and comprehensively discusses the application of nanocellulose membrane in microbial filtration, but the overall discussion needs to be more in-depth. There is a lack of comparative discussion, so it is not suitable to be published directly. I hope to continue to improve it.
- The differences of nanocellulose membranes prepared by different methods and different raw materials are not discussed, especially in terms of strength, porosity, virus filtration capacity and filtration efficiency.
- The similarities and differences between nanocellulose membrane and other existing filter membranes are not compared and analyzed, such as strength, filtration efficiency, ease of use, filtration pressure, etc.
- The filtration mechanism, such as adsorption mechanism or volume exclusion mechanism, should be further discussed.
- The functionalization of nanocellulose membrane is not discussed in detail. For example, is it the functionalization modification of nanocellulose membrane or the preparation of nanocellulose membrane after functionalization? What are the similarities and differences between the two, and what is the impact on the strength and filtration mechanism?
- In the discussion of virus filtration with nanocellulose membrane, several literatures were cited, focusing on the difference of membrane thickness. What are the other differences? What is the relationship between filtration capacity and thickness? What is the main effect of thickness? Porosity? Pore structure? Or membrane strength?
- Why are there two ways of reference in the article?
Author Response
Response to reviewer’s comments
Comment 1
The differences of nanocellulose membranes prepared by different methods and different raw materials are not discussed, especially in terms of strength, porosity, virus filtration capacity and filtration efficiency.
Response: Thank you for your comment. We had included a discussion on that matter (Page 5, line 158-164, line 166-172; Page 7: line 242-267)
Comment 2
The similarities and differences between nanocellulose membrane and other existing filter membranes are not compared and analyzed, such as strength, filtration efficiency, ease of use, filtration pressure, etc.
Response: Thank you for your comment, we had included a discussion about that matter (Page 8, line 269-294)
Comment 3
The filtration mechanism, such as adsorption mechanism or volume exclusion mechanism, should be further discussed.
Response: Thank you for your comment. We had improved discussion on adsorption mechanism or volume exclusion mechanism (Page 7; line 197-240)
Comment 4
The functionalization of nanocellulose membrane is not discussed in detail. For example, is it the functionalization modification of nanocellulose membrane or the preparation of nanocellulose membrane after functionalization? What are the similarities and differences between the two, and what is the impact on the strength and filtration mechanism?
Response: Thank you for your comment, we had improved a discussion on the functionalization of nanocellulose (Page 13, line 356-371)
Comment 5
In the discussion of virus filtration with nanocellulose membrane, several literatures were cited, focusing on the difference of membrane thickness. What are the other differences? What is the relationship between filtration capacity and thickness? What is the main effect of thickness? Porosity? Pore structure? Or membrane strength?
Response: Thank you for your comment. We had improved the discussion on virus filtration (Page 14, line 428-458)
Comment 6
Why are there two ways of reference in the article?
Response: Thank you for your comment. We had revised the reference style.
Reviewer 2 Report
The manuscript was well organized and reported as a useful review paper.
Author Response
Thank you for your comment.